# Factors Influencing the Concentration of Fecal Coliforms in Oysters in the River Blackwater Estuary, UK

**Styliani Florini** [1], **Esmaeil Shahsavari** [2],[*], **Tien Ngo** [2], **Arturo Aburto-Medina** [1,2], **David J. Smith** [1] and **Andrew S Ball** [1,2]

[1] Department of Biological Sciences, University of Essex, Colchester CO4 3SQ, UK;
stella.p.florini@gmail.com (S.F.); arturoaburto.medina@rmit.edu.au (A.A.-M.); djsmitc@essex.ac.uk (D.J.S.);
andy.ball@rmit.edu.au (A.S.B.)

[2] Centre for Environmental Sustainability and Remediation, School of Science, RMIT University, Victoria 3083,
Australia; chriswork0311@gmail.com

[*] Correspondence: esmaeil.shahsavari@rmit.edu.au

**Abstract:** Contamination of water systems can not only entail high risks to human health but can also result in economic losses due to closure of beaches and shellfish harvesting areas. Understanding the origin of fecal pollution at locations where shellfish are grown is essential in assessing associated health risks—as well as the determining actions necessary to remedy the problem. The aim of this work is to identify the species-specific source(s) of fecal contamination impacting waters overlying the shellfisheries in the Blackwater Estuary, East Anglia, UK. Over a twelve-month period, water samples were taken from above the oysters and from a variety of upstream points considered to be likely sources of fecal microorganism, together with oyster samples, and the number of fecal streptococci and *E. coli* were determined. Transition from low to high tide significantly decreased the concentration of fecal streptococci in waters overlying the oyster beds, indicative of a freshwater input of fecal pollution in oyster bed waters. In 12 months, the number of *E. coli* remained constant throughout, while fecal streptococci numbers were generally higher in the winter months. Analyses of upstream samples identified a sewage outfall to be the main source of *E. coli* to the oyster beds, with additional fecal streptococci from agricultural sources. The findings may assist in developing approaches for assessing the risks to shellfishery industries of various fecal inputs into an estuary, which could then help local governmental authorities address the problem.

**Keywords:** fecal contamination; oysters; shellfisheries; *E. coli*; fecal streptococci

## 1. Introduction

Outbreaks of enteric and other infectious diseases have been reported worldwide and have been attributed to bathing and eating molluskan shellfish from waters contaminated with fecal matter [1,2]. Contamination of water systems not only exact high risks to human health but also result in economic losses due to closure of beaches and shellfish harvesting areas. Waters contaminated with human feces are generally perceived as constituting a greater risk to human health than those contaminated with animal feces as they are more likely to contain human-specific enteric pathogens [3–5]. However, animal wastes are also known to contain a variety of enteric pathogens, some of which can cause disease in humans (e.g., species of *Campylobacter* and *Salmonella*, some toxigenic strains of *E. coli* and *Cryptosporidium* spp.) [3,6,7]. In the United States, Stelma and McCabe (1992) [8] demonstrated a link between human disease and consumption of shellfish harvested from water contaminated by only animal fecal wastes [7]. The sources of microbial contaminants found in freshwaters (such as

rivers, streams and estuaries) and coastal waters include: (1) discharges of untreated sewage or treated sewage effluent; (2) run-off from adjacent land areas, particularly land used for livestock farming; (3) recreational use; (4) stormwater drains; (5) industrial effluents; (6) fecal inputs from wild animals like birds; and (7) domestic animals such as dogs and cats [9–12].

Understanding the origin of fecal pollution at a particular location where shellfish are grown is essential in assessing associated health risks as well as the actions necessary to remedy the problem [13]. Moreover, management and mitigation of fecal pollution entering shellfish waters would be more cost-effective if the correct sources could be identified and apportioned [14]. In the European Union, controls intended to address these problems are exerted under the Shellfish Hygiene Directive 91/492/EEC [15] (Table 1) and these include classification of harvesting areas according to the degree of fecal pollution in shellfish flesh [16].

**Table 1.** Criteria for each of the categories and an indication of what treatment is required before mollusks can be placed on the market in accordance with the requirements in EC Regulation 854/2004, Annex II, Chapter II, A (EC 2004a).

| Category | Criteria | Treatment |
|---|---|---|
| Class A | Mollusks must contain less than 230 *E. coli* per 100 grams of flesh | Can be harvested for direct human consumption |
| Class B | 90% of sampled mollusks must contain less than 4600 *E. coli* per 100 grams of flesh; 10% of samples must not exceed 46,000 *E. coli* per 100 grams of flesh | Can go for human consumption after purification in an approved plant or after relaying in an approved Class A relaying area or after an EC approved heat treatment process |
| Class C | Mollusks must contain less than 46,000 *E. coli* per 100 grams of flesh | Can go for human consumption only after relaying for at least two months in an approved relaying area followed, where necessary, by treatment in a purification center or after an EC approved heat treatment process |
| Prohibited Area | Above 46,000 *E. coli* per 100 grams of flesh | Must not be harvested or offered for human consumption |

The River Blackwater Estuary is one of the largest estuaries in East Anglia, UK (Figure 1) and like many coastal areas represents a highly productive area [17], providing a rich resource for many species of wildlife, as well as for commercial shellfish cultivation. The intertidal areas, which consist of two main inter-related components, saltmarsh and mudflats, are particularly important in the estuarine ecosystem as well as providing an important natural sea defense [18]. The native oyster breeds throughout the Blackwater Estuary, especially in the outer reaches to the south of Mersea Island. The oysters are filter feeders and phytoplankton are their main source of food. Filter feeding animals consume large numbers of particles (any suspended material, including pathogenic bacteria or viruses) from the water indiscriminately. This ability to retain fecal micro-organisms makes shellfish a potentially useful tool for determining fecal contamination of surrounding waters [7]. The wide variety of fisheries, wildlife and recreational activities supported by the Blackwater Estuary requires a high water-quality standard. Currently the classification of shellfish in this area is B (Table 1); however recent reductions in the numbers of oysters have raised concerns about the possible increased levels of water contamination. The aim of this work was to identify the specific source(s) of fecal contamination impacting the shellfisheries in the Blackwater Estuary, which would then help local governmental authorities and the shellfisheries industry in finding an effective sustainable solution in maintaining oyster quality (Table 1). These research findings have implications to many oyster-growing estuaries which are exposed to various sources of fecal contamination.

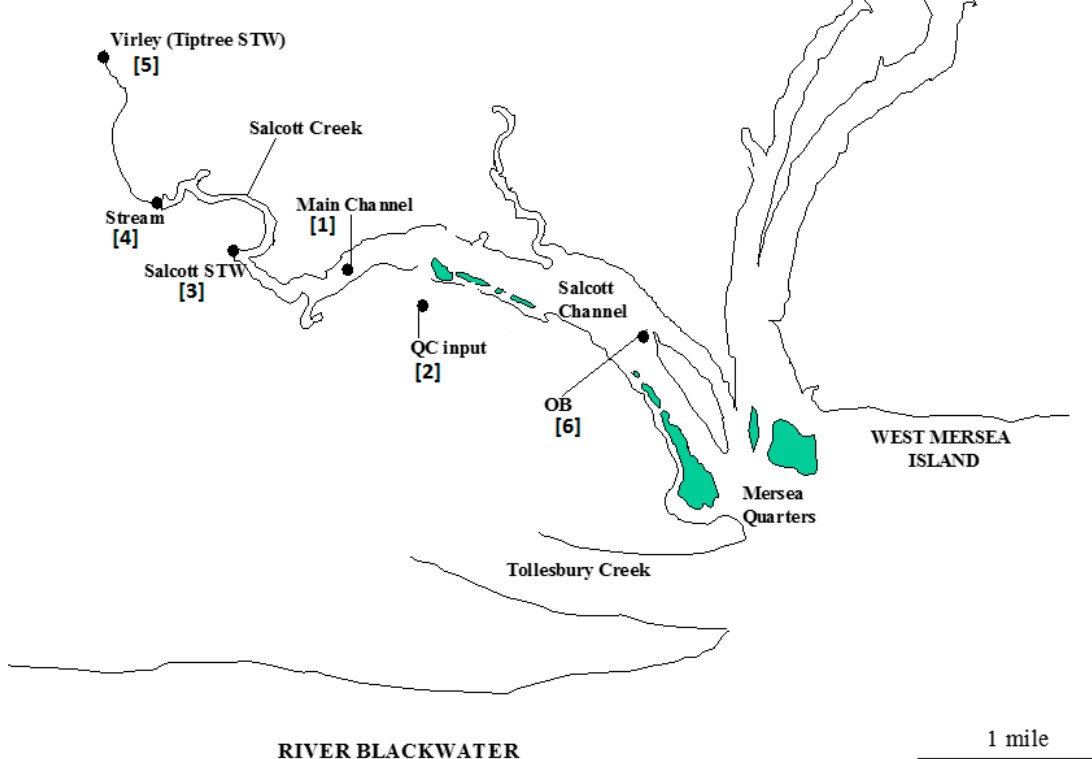

**Figure 1.** Map of Salcott Creek Channel, Essex UK showing the location of the 6 sampling sites. [1] Main Channel, [2] Quinces Corner, [3] Salcott Sewage Treatment Work Outlet Pipe, [4] Stream, [5] Virley (Tiptree Sewage treatment Works), [6] Oyster Beds. The green areas on the map show the location of small islands within the estuary.

## 2. Methods

### 2.1. Description of the Sampling Area

Salcott Creek/Channel is located to the west of West Mersea Island at the mouth of the Blackwater Estuary of the North Sea (Figure 1); it occupies a surface area of approximately 18 km². The different sampling sites used in this study are summarized in Table 2 and shown in Figure 1.

**Table 2.** The different sampling sites used in this study.

| SITE NO | SITE NAME | SOURCE | pH | Salinity | COMMENTS |
|---|---|---|---|---|---|
| 1 | Main Channel | Water | 7.2–8.5 | 0 | |
| 2 | Quinces Corner input (QC) | Water | 7.2–8.5 | 10–30 ppt on low tide and between 30 and 32 on high tide | Water originates from adjacent farmland (grazed by sheep) and also from surrounding fields where large flocks of Brent geese rest, especially during the winter period |
| 3 | Salcott STW (outlet pipe) | Water | 7.2–8.5 | 0 | Salcott Sewage treatment work (STW) outlet pipe. Salcott STW receives secondary treatment (conventional primary tanks with trickling filter and settlement tank as secondary treatment). |
| 4 | Stream | Water | 7.2–8.5 | 0 | Water from pastureland drains here via a tidal sluice |
| 5 | Virley (Tiptree STW) | Water | 7.2–8.5 | 0 | Water at this site originates from Tiptree STW. Sewage effluent from Tiptree STW is piped for approximately 3.5 km and then discharged at Virley Brook. |
| 6 | Oyster Beds (OB) | Water, oysters | 7.2–8.5 | 22–34 ppt during low tide and from 32–35 ppt during high tide | |

NB. Sites 2, 3, 4 and 5 constitute the main suspected inputs of fecal pollution in Salcott Creek.

## 2.2. The Effect of Tidal Regime on E. coli and FS Numbers in Salcott Creek

To investigate the effect of tidal state on bacterial concentrations in Salcott Creek, water was collected above the Oyster Beds (OB) every hour between high and low tide and between low and high tide in Autumn and Summer. For all samples, temperature was measured *in situ* and salinity and pH were measured in the laboratory within 2–4 h of the sample being taken. Finally, microbiological analyses were carried out for all samples using the MTP technique (described below).

## 2.3. The Effect of Distance from the Main Sewage Source on the Numbers of E. coli and FS in Salcott Creek

Water (effluent) was initially collected from the STW discharge pipe in Salcott Village (site 3) and then approximately 1.5 km (point 1), 2 km (point 2), 3 km (point 3), 3.5 km (point 4) and 4 km (point 5) further downstream along Salcott Creek. The last sampling point (6) was at the OB (site 6). Both *E. coli* and FS concentrations were estimated for all samples using the MTP method described below. Sampling took place during low tide (±1.1 m relative to mean sea level), and the pH and salinity of all water samples were measured in the laboratory on the same day of sampling.

## 2.4. Twelve-Month Microbiological Study

Water samples were collected bimonthly over a 12-month period. All samples were collected at low tide. Water samples were collected in 250 mL, pre-autoclaved bottles, with three replicates per site. When sampling with the boat, water was collected by immersing the bottles approximately 25 cm below the water surface. Oysters were collected from the oyster beds and transported in a plastic bag containing ice packs. All samples were transported on ice to the laboratory and analyzed within 24 h of collection using the MTP method described below.

## 2.5. Bacterial Analysis of Water

An aliquot (1 mL) of each water sample was mixed with 9 mL of sterile saline water. A series of 10-fold dilutions (up to $10^{-4}$ for sediments and $10^{-3}$ for water) were prepared. Water was analyzed using the MTP method (Pourcher et al., 1991). Briefly, A-1 medium used for the detection of *E. coli*

was amended with 4-methyl-umbelliferyl-β-ᴅ-glucuronide (MUG) while for the enumeration of FS, methylumbelliferyl-β-ᴅ-glucoside (MUD) was used (Pourcher et al., 1991). The concentrations of *E. coli* and FS were estimated according to the most probable number calculation with 8 wells and 5 tubes per dilution. The results were expressed as number of bacteria per 100 mL of water.

### 2.6. Bacterial Analysis of Oysters

Oysters (n = 6) were cleaned by scraping, scrubbing and washing under cold running water and then allowed to drain on clean paper towels. They were then opened using a sterile (pre-autoclaved) oyster knife and the meat and internal fluid of oysters was pooled, and an aliquot (10 mL) homogenized in a Waring blender using two volumes of sterile 0.1% (wt/vol) peptone water. The homogenate (30 mL) was mixed with peptone water (70 mL), resulting in a $10^{-1}$ dilution. A series of ten-fold dilutions up to $10^{-3}$ were then prepared. Finally, analyses were carried out in triplicate. A volume of 1 mL of each dilution was added to each of five A-1 Durham fermentation tubes and to five azide dextrose broth tubes for the enumeration of *E. coli* and FS respectively. After inoculation, tubes were incubated for 36 h at 44.5 °C for *E. coli* and at 37 °C for FS. Following incubation, gas production coupled with growth in each EC broth tube was considered a positive *E. coli* reaction. FS positive tubes and wells were confirmed by streaking onto Bile Aesculin agar. For each bacterial indicator the number of positive tubes for each dilution was recorded and concentrations of bacteria were calculated per $100 \times g$ of oyster flesh using MPN tables using MPN tables.

### 2.7. Abiotic Measurements

Abiotic measurements such as temperature, pH and salinity were carried out prior to bacterial analysis of water samples. Water temperature (°C) was measured in situ using a handheld digital thermometer. pH was measured using a digital pH meter which was calibrated to pH 7.0. Salinity (ppt) was measured using a handheld 'Leica' temperature compensated refractometer.

### 2.8. Statistical Analysis

MPN returns discontinuous data and therefore non-parametric statistical analysis was used for all the MPN data. For comparisons between two samples a Mann–Whitney U test was used and for multiple sample comparisons a Kruskal–Wallis test was used. To determine any association between variables, Spearman's correlation tests were carried out. All statistical tests were performed using SigmaStat.

## 3. Results and Discussion

### 3.1. Assessment of the Tidal Effects on E. coli and FS Numbers in Waters Overlying the Oyster Bed (OB)

To determine whether the number of fecal indicators present in waters above the oyster beds (OB) varied with tides, samples were taken throughout the tidal cycle (Figure 2). Both *E. coli* and FS concentrations reached their maximum values during low tide. *E. coli* numbers ranged from 0 to 208 cells 100 mL$^{-1}$ and FS from 150 to 4483 cells 100 mL$^{-1}$ on high tide and low tide respectively (Figure 2A). There was a significant decrease in both *E. coli* ($p < 0.001$) and FS ($p < 0.05$) numbers from low to high tide (Figure 2A). Transition from high to low tide (Figure 2B) had the opposite effect; numbers of FS dropped significantly from 700 to 33 cells 100 mL$^{-1}$ of water (Figure 2B) ($p < 0.01$). The numbers of *E. coli* were constantly zero from low tide until high tide (Figure 2A). Transition from low to high tide significantly decreased the concentration of FS in waters overlying the oyster beds (Figure 2A). This could be due to significant dilution of bacterial indicators resulting from the mixing of seawater with freshwater as the tide goes in. However, transition from high to low tide had the opposite effect on the concentrations of both *E. coli* and FS (Figure 2B). The increase in the levels of both fecal indicators was statistically significant and indicative of a freshwater input of fecal pollution in OB waters. This result correlates well with the findings of Erkenbrecher (1981) [19] who determined

that in Lynnhaven Estuary (a sub-estuary of the lower Chesapeake Bay), sites with higher salinity water showed lower overall bacterial densities than did the headwater sites, where freshwater runoff and decreased tidal action were characteristic. Finally, decreased bacterial concentrations during high tide could be due to both dilution by seawater and/or increases in salinity during high water. Salinity can affect fecal bacterial viability, with high or rapidly changing salt concentrations increasing cell inactivation [20]. Zaccone et al. (2005) [21] also reported higher bacterial counts were associated with lower salinity values as well higher ammonia concentrations.

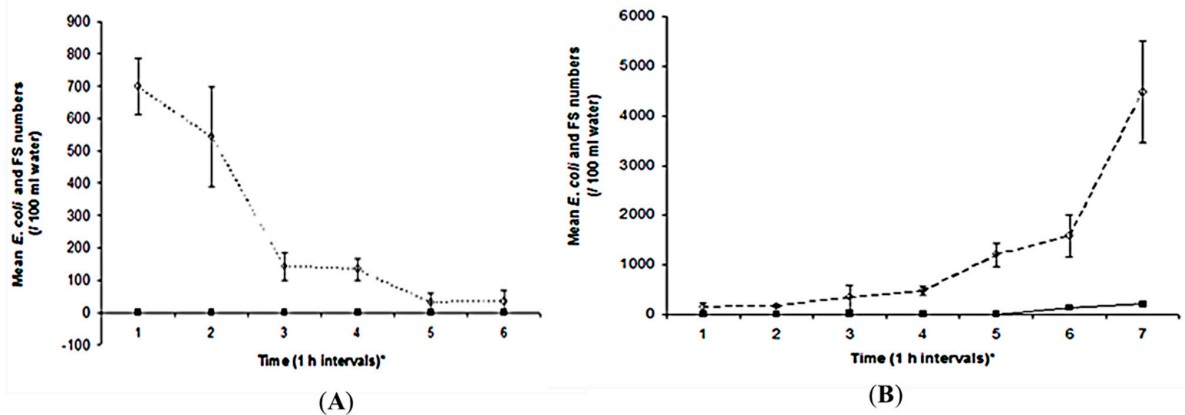

**Figure 2.** Changes occurring in the concentrations of *E. coli* (solid squares and line) and FS (fecal streptococci, empty circles and broken line) from **A**. low to high tide and **B**. high to low tide where 1 = water collected 30 min before high (**A**) or low (**B**) water and 2–7 = water collected at 1 h intervals between tides. Means ± SE (n = 3) are presented.

### 3.2. Seasonal Variations in Microbiological Quality of Water and Oyster Samples Collected from the Oyster Beds

Having established that the main source of fecal contamination at the oyster beds were likely to come from upstream, seasonal variations in the number of FS and *E. coli*, both in the oysters themselves and in the waters above the beds were investigated. Over a 12-month period samples of oysters and the overlying water were collected together with samples for the various potential inputs of fecal organisms shown in Figure 1, to assess what were the major factors influencing the number of fecal indicator organisms in the oysters. At the oyster beds *E. coli* concentrations ranged from 0 to 367 cells 100 mL$^{-1}$, while FS concentrations were statistically higher ($p < 0.001$), varying between 0 and 13,500 cells 100 mL$^{-1}$. The highest FS counts were recorded during the winter months (Figure 3A). There was no seasonal variation observed for *E. coli* counts; this was confirmed by the results of a Kruskal–Wallis multi-comparison test, which indicated that there was no significant difference ($p > 0.05$) in *E. coli* numbers between the four different seasons. However, in the case of FS a seasonal variation was observed. Numbers of FS were high in winter, autumn and spring compared with summer (see Figure 3A). There was a significant difference ($p = 0.005$) in the concentrations of FS between the four seasons. The results of pairwise multiple comparison tests indicated that the above difference was due to significant differences in FS numbers between winter and summer (q = 2.987, $p < 0.05$) and between winter and autumn (q = 3.047, $p < 0.05$). Chandran and Hatha, (2003) [22] reported that sunlight is one of the most important inactivating factors for the survival of bacteria in estuarine waters and this may be the reason for lower FS counts during the winter months. This indicates that fecal coliforms and *E. coli* levels are elevated during the monsoon months (low temperatures) and less prevalent during the pre- and post-monsoon months (high temperatures). Similar conclusions were reached by Kucuksezgin et al. (2010) [23], who found more fecal coliforms in clams during the colder winter-spring period in Izmir Bay, Turkey.

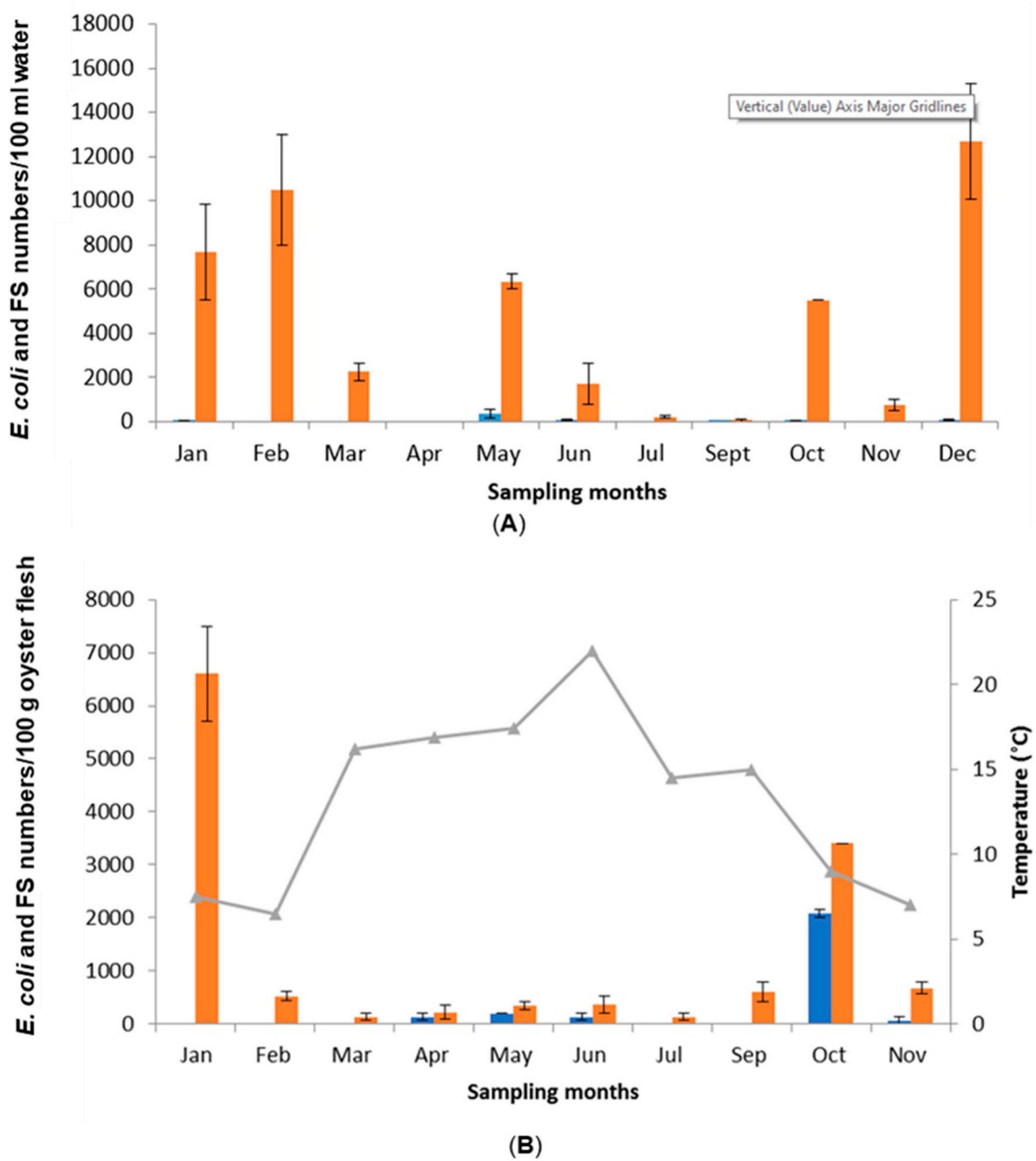

**Figure 3.** (**A**). Concentrations of *E. coli* (blue bars) and FS (orange bars) in water samples collected from the oyster beds over a period of 12 months. (**B**). Concentrations of *E. coli* (blue bars) and FS (orange bars) of oysters collected from the oyster beds over a period of 12 months Results shown are means ± SE (n = 3). Where no bar is present counts = 0. Gray line shows the water temperature (°C) over the 12-month period.

Concentrations of *E. coli* in oysters collected from Salcott Creek ranged from 0 cells 100 g$^{-1}$ oyster flesh (on many sampling occasions) to 2083 cells 100 g$^{-1}$ oyster flesh in October. Counts of FS varied between 133 cells 100 g$^{-1}$ oyster flesh in March and July, and 6600 cells 100 g$^{-1}$ oyster flesh in January (Figure 3B). There was a statistically significant difference (*p* = 0.007) between *E. coli* numbers and sampling months and also between FS counts and sampling months (*p* = 0.005). The former was due to the fact that on many sampling occasions there were 0 counts for *E. coli*, while the latter was due to the high counts of FS recorded in October and January compared with the rest of the sampling months where FS were present at much lower concentrations (Figure 2B). According to the EC shellfish harvesting directive 91/492/EEC, all oysters (except those collected in October) fall

into the Class A category (<230 *E. coli* per 100 g oyster flesh). Oysters collected in October were class B (<4600 *E. coli* per 100 g oyster flesh) (Table 1). The results largely reflect the results obtained for FS in water (Figure 2A). It has been reported that shellfish accumulate coliforms from the shellfish growing waters and then maintain these levels for long periods [24,25]. In this study, there were no significant differences (*p* > 0.05) in either *E. coli* or FS numbers between water and oysters sampled on the same dates suggesting that uptake by oysters was linked to the number of FS or *E. coli* present in the overlying waters.

### 3.3. Comparison of Fecal Indicator Numbers between the Different Inputs

To understand the potential sources of input of FS and *E. coli*, samples were collected from the various potential inputs site (Figure 1) over the same 12-month period. During the 12-month period, the highest mean *E. coli* counts were recorded in waters collected from Salcott STW and the lowest in QC waters (Figure 4A,B, respectively). Intermediate *E. coli* numbers were recorded in Virley and Stream samples (Figure 4C,D respectively). There was a significant difference in *E. coli* counts between the different inputs (*p* < 0.001). Salcott STW was found to be the major contributor of *E. coli* loadings in the waters above the Oyster Bed followed by Virley (Figure 4B and Figure 4D, respectively). However, in cases where *E. coli* numbers between Salcott STW and Virley are similar, then Virley was considered as the main source of *E. coli* present in Salcott Creek as it discharges approximately 10 times more effluent than the Salcott sewage pipe.

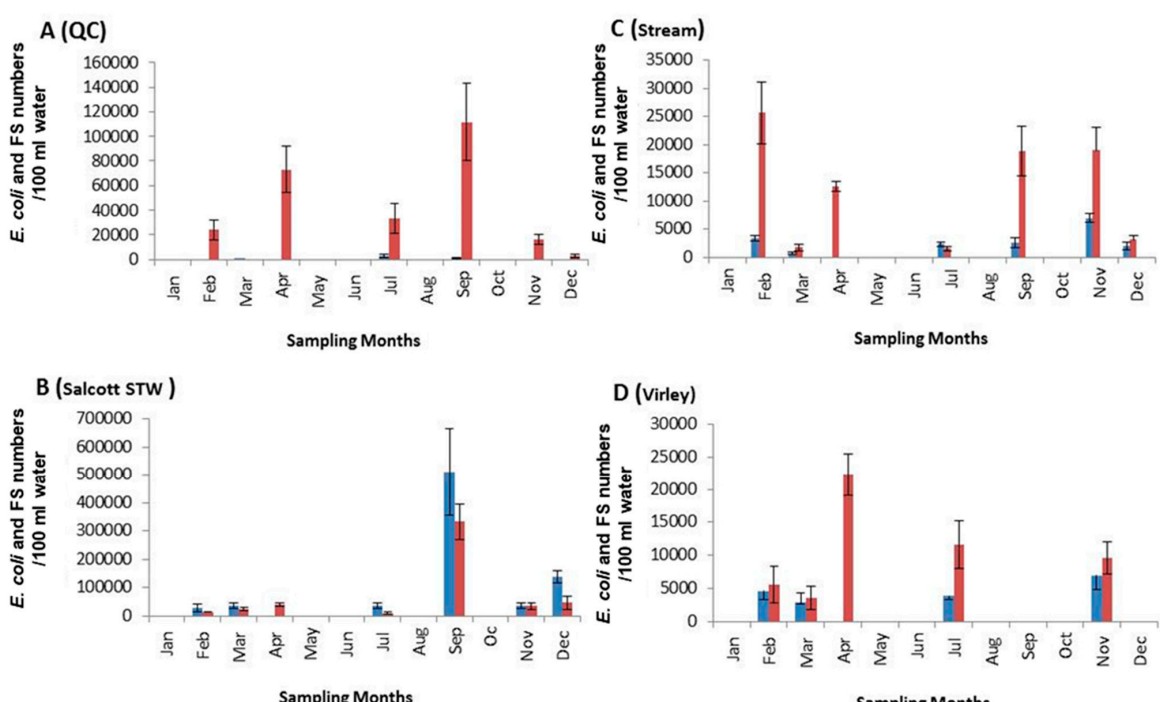

**Figure 4.** Concentrations of *E. coli* (Blue bars) and FS (Red bars) in water samples collected from QC (**A**), Salcott STW (**B**), Stream (**C**) and Virley (**D**) input over 12 months. Means ± SE (n = 3). Where no bar is present counts = 0.

The highest mean FS counts were recorded in Salcott STW and QC waters (Figure 4B and Figure 4A). Therefore, these two inputs represent the major contributors of FS loadings in Salcott Creek/Channel. FS numbers at Virley and the Stream were also high but still lower than those at Salcott STW and QC (Figure 4D). Weak, statistically insignificant correlations between FS counts at the OB and Salcott STW, stream and Virley (sites 4, 5 and 6) were observed but there was a strong positive correlation between FS numbers at the OB and QC ($R^2$ = 0.703, *p* < 0.05, n = 21). A significant positive correlation between *E. coli* counts at the OB and Virley ($R^2$ = 0.700, *p* < 0.05, n = 39) was

detected together with a weak statistically insignificant correlation between *E. coli* numbers at the OB and the rest of the inputs. In a previous study Kay et al. (2008) [26] conducted field surveys to examine the effects of sewage contamination from storm overflow effluent on *E. coli* and fecal coliform concentrations in the flesh of wild mussels (*Mytilus edulis*). *E. coli* (and fecal coliform concentrations) in shellfish increased rapidly after discharge, with *E. coli* concentrations exceeded the European shellfish hygiene class C limit of 46,000 per 100 $g^{-1}$ and decayed during subsequent discharge-free periods. Clements et al. (2015) [27] reported heterogeneity in the spatial distribution of bacteria at an intertidal of shellfish bed.

### 3.4. The Effect of Distance from the Sewage Source on the Numbers of E. coli and FS in Salcott Creek

To investigate whether there were any additional inputs of fecal matter in Salcott Creek, the effect of distance from the main sewage source (Salcott STW) on the distribution of *E. coli* and FS was examined. The concentrations of both fecal indicators decreased with increasing distance from Salcott STW discharge point most likely due to dilution and die-off (Figure 5). This decrease was more rapid for *E. coli* compared to FS and this was probably related to the higher survival of the later in the water column [12,28,29]. However, between sampling points 3 and 4 (3 km and 3.5 km downstream from Salcott STW respectively) there was a sudden increase of almost 2500 cells in the concentration of FS but not *E. coli* (Figure 5). This increase suggests an additional fecal input, which could be attributed to agricultural run-off since there was high rainfall on the day of sampling and prior to sampling. Salcott Creek is surrounded by agricultural land grazed by sheep and the increase in the numbers of FS only may be due to their predominance in animal feces combined with their higher survival [3,29–31]. Regression analysis also was performed; a weak though not significant correlation between *E. coli* and FS with distance in Salcott Creek was observed ($p = 0.074$).

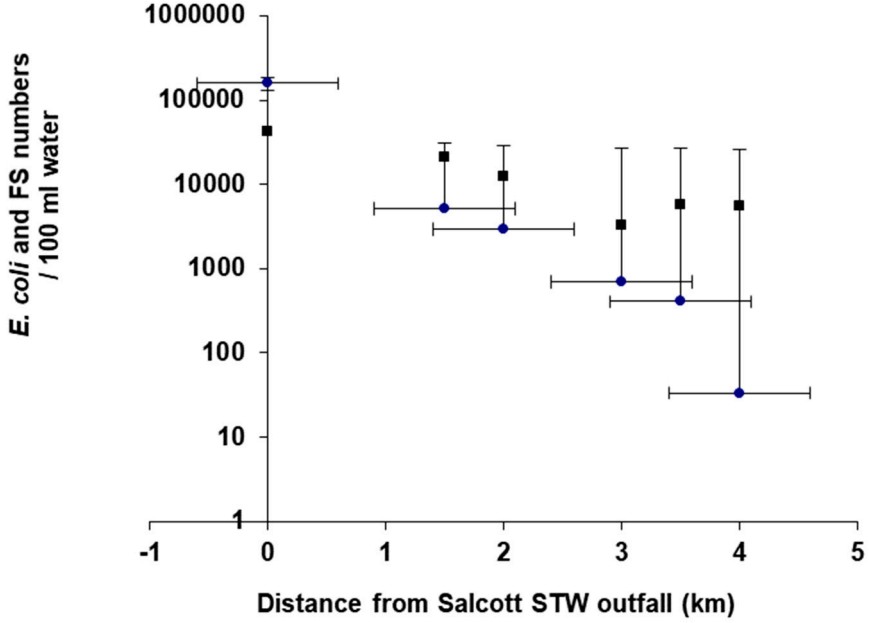

**Figure 5.** The effect of distance from Salcott STW on the distribution of *E. coli* (circles) and FS (squares) along Salcott Creek. Means ± SE (n = 3).

We have identified that both inputs; a point source from local wastewater treatment works and a non-point source, grazing livestock both significantly impact water quality in the estuary. Focused management outcomes such as the implementation of improved wastewater treatment processes together with a reduction in livestock grazing will significantly reduce the bacterial load in the oyster bed overlying water, caused by point and non-point sources, thereby preserving the quality the oysters. A recent paper by Wu (2019) [32] reported on the linking of landscape patterns to sources of water

contamination through geospatial and Bayesian approaches. Such approaches can be applied in future research examining the implications of land management on shellfishery industries in estuaries.

## 4. Conclusions

Understanding the origin of fecal pollution at a particular location where shellfish are grown is essential in assessing associated health risks as well as the determining actions necessary to remedy the problem. Here we examined the abundance of fecal streptococci and *E. coli* in both oysters and overlying water in a coastal estuary in the UK. The transition from low to high water significantly decreased the concentration of fecal streptococci in waters overlying the oyster beds, indicating a freshwater input of fecal pollution in oyster bed waters. The concentrations of fecal indicators in oysters and overlying water were followed over 12 months. Generally, the number of *E. coli* remained constant throughout while fecal streptococci numbers were generally higher in the winter months. Analysis of upstream samples identified a sewage outfall to be the main source of *E. coli* to the oyster beds, with additional fecal streptococci from agricultural sources. Understanding the origin of fecal pollution at a particular location where shellfish are grown is essential in assessing associated health risks as well as the actions necessary to remedy the problem. The approach taken in this study can be applied to other estuaries where shellfish are grown and will lead to the implementation of informed management practices in surrounding areas which will lead to a reduction in fecal coliform contamination through both point and non-point sources.

**Author Contributions:** Conceptualization, D.J.S. and A.S.B.; Data curation, S.F.; Formal analysis, E.S., T.N. and A.A.-M.; Investigation, S.F., A.S.B. and D.J.S.; Methodology, S.F.; Project administration, A.S.B. and D.J.S..; Supervision, A.S.B.; Validation, E.S., T.N. and D.J.S.; Visualization, D.J.S.; Writing—original draft, S.F. and A.S.B.; Writing—review & editing, E.S., T.N., A.A.-M. and A.S.B. All authors have read and agreed to the published version of the manuscript.

**Funding:** This research received no external funding.

**Conflicts of Interest:** The authors declare no conflict of interest.

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
