# Peer review of "Factors Influencing the Concentration of Fecal Coliforms in Oysters in the River Blackwater Estuary, UK"

_water, doi:10.3390/w12041086_

Round 1

Reviewer 1 Report

Comments attached.

Author Response

L17 There are some grammar issues throughout the paper starting here;

Answer: We apologies for the oversight. These errors have been corrected through careful proof reading by a native English speaker.

L72 Not immediately clear of the implications of the water being class B. Maybe an explanation in more general terms would be more clear. Or maybe a reference to table 1 would be useful.

Answer: Table1 has been added now, which clarifies the importance of the Class B status.

Fig 1 What are the green patches on the map?

Answer: The green areas on the map show the location of small islands within the estuary. We have clarified this on the Figure legend.

Fig 2 A log y-axis would be more clear here. Parallel structure between the two panels is needed. The two panels are also of different sizes.

Answer: The error has been fixed. We apologise for the error in stating log. This was incorrect and confusing.

L165-166 Could the decreased numbers of indicators also be due to dilution?

Answer: Thank you for you input. Indeed, dilution by seawater is expected when freshwater is the source of the bacterial load, as stated earlier (line 152),. To clarify further, we have amended the sentence accordingly to read “Finally, decreased bacterial concentrations during high tide could be due to both dilution by sweater and/or increases in salinity during high water.”.  

Fig 3 A legend would be very useful in interpreting the bar colors. What is “Vertical (Value) Axis Major Gridlines” doing there? What does that mean? The y-axis indicates the data have been log-transformed prior to plotting, but y-axis tick marks indicate otherwise. I could be colorblind but the colors in the plot do not match the colors described in the legend (e.g., “Green line”?)’

Answer: We apologise for the oversight. The error has been fixed in Figure 3.

L236 Careful between correlation and linear regression. Which is being performed? Does “rs” stand for R2?

Answer: Many thanks for pointing out the distinction. We have amended the text throughout to R2.

Major comments:

There was very little attention given to displaying the data, which makes the paper very difficult to read and evaluate conclusions. Authors appear to be confused by log-transformation of the data and by log-transformed axes. In this vein the statistical methods are unclear; where the data log-transformed before correlation analysis?

Answer: Please accept our apologies for the confusion regarding the incorrect labelling of the y axis in Figures 2, 3a and 4. These axes labels were left from a previous version of the Figures. The data presented in the figures are not log transformed. Consequently, for statistical analyses we do not use log transformed data as non-parametric statistical analyses were used. Only Fig 5 shows log data as it presents a clear correlation with distance. We apologize for the oversight and have carefully corrected the manuscript so as not to cause confusion.

Not enough discussion is given to which indicator to believe. They sort of tell different stories of contamination at some points, which leaves authorities who are supposedly using this information to fix the problem still guessing at which management actions to take.

Answer: Many thanks for your reflection. To address this point we have added the following text at the end of Section 3.4. “We have identified that both inputs, a point source from local wastewater treatment works and a non-point source, grazing livestock both significantly impact water quality in the estuary. Focused management outcomes such as the implementation of improved wastewater treatment processes together with a reduction in livestock grazing will significantly reduce the bacterial load in the oyster bed overlying water, caused by point and non-point sources, thereby preserving the quality the oysters. A recent paper by Wu (2019) [29] reported on the linking of landscape patterns to sources of water contamination through geospatial and Bayesian approaches. Such approaches can be applied in future research examining the implications of land management on shellfishery industries in estuaries.

To further clarify the outcome and significance of this work in the Conclusion section we have also added “The approach taken in this study can be applied to other estuaries where shellfish are grown and will lead to the implementation of informed management practices in surrounding areas which will lead to a reduction in faecal coliform contamination through both point and non-point sources”. We believe this addresses the reviewers' concern.

Reviewer 2 Report

Overall, the manuscript was well written. However, I have some minor comments:

  1. In the abstract and conclusion, the writing can be concise. In addition, it may be better to show some data instead of just describing the findings.
  2. In the method, the statistical analysis can be rigorous. For example, when you exam the effect of distance from the sewage source on the numbers of E. coli and FS, it is better to do regression analysis.
  3. Can you exam how land use/land cover influences the sources of microbial contamination? Recommend to read "Wu, J., 2019. Linking landscape patterns to sources of water contamination: Implications for tracking fecal contaminants with geospatial and Bayesian approaches. Science of The Total Environment, 650, pp.1149-1157".
  4. Line 96-97, can you clarify what are 'low water' and 'high water'?

Author Response

1. In the abstract and conclusion, the writing can be concise. In addition, it may be better to show some data instead of just describing the findings.

Answer: Throughout revisions we have considered the comment and have tried to be more concise in our writing and have included an extra Table to help us report the significance of the work. We thank the reviewer for the comment and hope that our changes have made the report more concise with the key findings highlighted.

2. In the method, the statistical analysis can be rigorous. For example, when you exam the effect of distance from the sewage source on the numbers of E. coli and FS, it is better to do regression analysis.

Answer: The authors are grateful for the comments. I believe our error in incorrectly using the term Log in Figure 2, 3a and 4 have caused some confusion as we reported above. We apologies for this. Our statistical expert Prof. Smith was consulted throughout the project in both experimental design and statistical analysis.

In the Methods section we report on the statistical analysis used “MPN returns discontinuous data and therefore non-parametric statistical analysis was used for all the MPN data. For comparisons between two samples a Mann-Whitney U test was used and for multiple sample comparisons a Kruskal-Wallis test was used. In order to determine any association between variables, Spearman’s correlation tests were carried out. All statistical tests were performed using SigmaStat”. Regarding Regression Analysis for Figure 5, we thank you for your comments. We have carried out analysis for this Figure and have now included the outcome in the text.

3. Can you exam how land use/land cover influences the sources of microbial contamination? Recommend to read "Wu, J., 2019. Linking landscape patterns to sources of water contamination: Implications for tracking fecal contaminants with geospatial and Bayesian approaches. Science of The Total Environment650, pp.1149-1157".

Answer: Thank you very much for your comment and useful paper. We have now included a paragraph at the end of Section 3.4 on the implications of the findings and the approach, making reference to the above paper which has now been included in the papers cited. The section reads “A recent paper by Wu (2019) [32] reported on the linking of landscape patterns to sources of water contamination through geospatial and Bayesian approaches. Such approaches can be applied in future research examining the implications of land management on shellfishery industries in estuaries”. We hope this addition further discusses the implications of the current work and suggests future research.

4. Line 96-97, can you clarify what are 'low water' and 'high water'?

Answer: We apologise for the oversight. Low and high water are in fact low and high tide3. We have replaced “water” with “tide” throughout.

Reviewer 3 Report

This manuscript describes " factors influencing the concentration of fecal coliforms in oysters in the River Blackwater Estuary, UK". The comments are below.

  1. In conclusion section, add which factor is major source of fecal coliform concentration.

2. In Figure 2, Insert different Y-axis. Authors need to make another y-axis for FS number. The scale of FS is too low.

3. Add and explain non-point source effect and how non-point source shoule be managed.

4. Add standard deviation in Figure 5.

5. If possible, write effect of non-point source to the number of E-coli and FS in this study.

6. If possible, update the references with recent papers.

Author Response

1. In conclusion section, add which factor is major source of fecal coliform concentration.

Answer: We thank the reviewer for their comment. We have addressed this point. The final sentences of the conclusion now read “Analysis of upstream samples identified a sewage outfall to be the main source of E. coli to the oyster beds, with additional faecal streptococci from agricultural sources. Understanding the origin of faecal pollution at a particular location where shellfish are grown is essential in assessing associated health risks as well as the actions necessary to remedy the problem. The approach taken in this study can be applied to other estuaries where shellfish are grown and will lead to the implementation of informed management practices in surrounding areas which will lead to a reduction in faecal coliform contamination through both point and non-point sources.”.

2. In Figure 2, Insert different Y-axis. Authors need to make another y-axis for FS number. The scale of FS is too low.

Answer: We apologise for our error. The error has been remedied.

3. Add and explain non-point source effect and how non-point source should be managed.

Answer: We thank the reviewer for their helpful comment. To address this point we have added the following paragraph at the end of Section 3.4.

We have identified that both inputs, a point source from local wastewater treatment works and a non-point source, grazing livestock both significantly impact water quality in the estuary. Focused management outcomes such as the implementation of improved wastewater treatment processes together with a reduction in livestock grazing will significantly reduce the bacterial load in the oyster bed overlying water, caused by point and non-point sources, thereby preserving the quality the oysters. A recent paper by Wu (2019) [29] reported on the linking of landscape patterns to sources of water contamination through geospatial and Bayesian approaches. Such approaches can be applied in future research examining the implications of land management on shellfishery industries in estuaries.

4. Add standard deviation in Figure 5.

Answer: We thank the reviewer for pointing out our error. We have addressed this point through the addition of standard deviation and regression analysis in Figure 5.

5. If possible, write effect of non-point source to the number of E-coli and FS in this study.

6. If possible, update the references with recent papers.

Answer: We thank the reviewer for pointing this additional, recent references have been added throughout the manuscript.

Round 2

Reviewer 2 Report

The authors have addressed my concerns.

Reviewer 3 Report

Once this manuscript is accepted, please edit Fig. 5 again.

The bar for the standard deviation should be revised. The bar should be in y-axis form.